# Platelet Counting: Ugly Traps and Good Advice. Proposals from the French-Speaking Cellular Hematology Group (GFHC)

**DOI:** 10.3390/jcm9030808

**Published:** 2020-03-16

**Authors:** Véronique Baccini, Franck Geneviève, Hugues Jacqmin, Bernard Chatelain, Sandrine Girard, Soraya Wuilleme, Aurélie Vedrenne, Eric Guiheneuf, Marie Toussaint-Hacquard, Fanny Everaere, Michel Soulard, Jean-François Lesesve, Valérie Bardet

**Affiliations:** 1Laboratoire d’hématologie, CHU de la Guadeloupe, INSERM UMR S_1134, 97159 Pointe-à-Pitre, France; 2Fédération Hospitalo-Universitaire ‘Grand Ouest Against Leukemia’ (FHU GOAL), 49033 Angers, France; frgenevieve@chu-angers.fr; 3Université Catholique de Louvain, CHU UCL Namur, Laboratoire d’hématologie, Namur Thrombosis and Hemostasis Center, 5530 Yvoir, Belgium; jacqmin.hugues@gmail.com (H.J.); bernard.chatelain@gmail.com (B.C.); 4Hospices Civils de Lyon, Centre de biologie et pathologie Est, Service d’hématologie biologique, 69500 Bron, France; sandrine.girard@chu-lyon.fr; 5Laboratoire d’Hématologie, Institut de Biologie, CHU de Nantes; 44093 Nantes CEDEX, France; soraya.wuilleme@chu-nantes.fr; 6Service de biologie clinique, Hôpital Foch, 92150 Suresnes, France; a.vedrenne@hopital-foch.com; 7Service d’Hématologie Biologique, CHU Amiens-Picardie, 80054 Amiens CEDEX, France; guiheneuf.eric@chu-amiens.fr; 8Hématologie Biologique, CHRU Nancy, 54511 Vandoeuvre, France; m.toussaint-hacquard@chru-nancy.fr (M.T.-H.); jf.lesesve@chru-nancy.fr (J.-F.L.); 9Audolys Biologie, 62219 Longuenesse, France; jfeveraere@audolys.fr; 10Plateau technique d’hématologie, Laboratoire Biogroup, 92300 Levallois-Perret, France; m.soulard@biogroup-lcd.fr; 11Service d’Hématologie-Immunologie-Transfusion, CHU Ambroise Paré, INSERM UMR 1184, AP-HP, Université Paris Saclay, 92100 Boulogne-Billancourt, France; valerie.bardet@aphp.fr

**Keywords:** platelet count, thrombocytopenia, thrombocytosis, proposals

## Abstract

Despite the ongoing development of automated hematology analyzers to optimize complete blood count results, platelet count still suffers from pre-analytical or analytical pitfalls, including EDTA-induced pseudothrombocytopenia. Although most of these interferences are widely known, laboratory practices remain highly heterogeneous. In order to harmonize and standardize cellular hematology practices, the French-speaking Cellular Hematology Group (GFHC) wants to focus on interferences that could affect the platelet count and to detail the verification steps with minimal recommendations, taking into account the different technologies employed nowadays. The conclusions of the GFHC presented here met with a "strong professional agreement" and are explained with their rationale to define the course of actions, in case thrombocytopenia or thrombocytosis is detected. They are proposed as minimum recommendations to be used by each specialist in laboratory medicine who remains free to use more restrictive guidelines based on the patient’s condition.

## 1. Introduction 

Platelets are the smallest cellular elements of the blood. They are anucleate fragments of the cytoplasm of bone marrow megakaryocytes and are essential for hemostasis. 

Platelet count is an essential examination in patient management and an important diagnostic tool in hemorrhagic disorders. It is also strongly recommended in the follow up of potentially thrombocytopenia-inducing treatments, such as anti-cancer chemotherapy and anticoagulants, e.g., heparin. Platelet count must be performed in an accurate and reliable way. The hematology analyzers currently available on the market use different technologies to count the platelets: impedance, optical methods (light diffraction or fluorescence techniques), and immunofluorescence techniques using monoclonal antibodies directed against glycoproteins found on the surface membrane of platelets. In addition, the platelet count is subjected to variations related to artifacts well known by laboratory biologists and must be ruled out before validation. The pre-analytical phase is also crucial, especially when sodium citrate anticoagulant is used instead of EDTA. An abnormal or doubtful platelet count must, therefore, be carefully checked with the help of well-established decision trees. Analytical pitfalls inherent to the technology used by the various hematology analyzers must be well known by users in order to ensure the right result. A brief overview of the currently available techniques is given below before discussing the interferences that could impact platelet count. Then, we have detailed our proposals facing thrombocytopenia and thrombocytosis. In both cases, we have explained how to check the platelet count and how to report its result. This document was produced by a working group on behalf of the French-speaking Cellular Hematology Group (GFHC); the group was formed by 13 experts in cell hematology, adult or pediatric settings, from university hospitals, general hospitals, or private practice. 

## 2. Platelet Count

### 2.1. Different Counting Techniques

#### 2.1.1. Microscopic Methods

The manual phase contrast microscopy method was described by Brecher in 1953 [1] and has been the only reference method for platelet counting for a long time. This laborious technique suffers from imprecision and poor reliability: interobserver CV (coefficient of variation) is 10–25% for normal samples and goes up to 40% for thrombocytopenic samples. However, this method is still used in clinical laboratories when atypical platelets (e.g., giant platelets) are present in the sample [2,3].

An alternative method for manual platelet counting is the counting performed on peripheral blood smear [4]. It provides a simple double platform platelet count based on the ratio of observed platelets to red blood cells (RBC), the enumeration of which is obtained with the automated count [5]. A more recent application of the old method used the ratio of platelets to leukocyte count [6,7].

The digital microscope can provide a reliable platelet count via direct identification by the cytologist on a wide high-power field. This method is useful to avoid artifacts generating over or underestimation of automated platelet counts and has been reported to be more efficient and more accurate than an alternative microscopic method. This is mainly due to the high-resolution scanning of the defined area of the automated blood smear preparation [8].

#### 2.1.2. Immunoplatelet Counting

Immunological platelet counting is the reference method [9,10] endorsed by ICSH (International Council for Standardization in Haematology) and ISLH (International Society for Laboratory Hematology) in the process of whole blood calibration of automated hematology analyzers [11]. Platelets are labeled by FITC-conjugated monoclonal antibodies against two distinct epitopes of the integrin αIIbβ3 (CD41 and CD61) and analyzed with a flow cytometer to calculate the ratio of fluorescent platelets to RBC contained in the same preparation (Figure 1). The reference platelet count is calculated from this ratio, and the RBC count is obtained by an impedance-based automatic counter. This method is particularly useful when thrombocytopenia is severe (< 20 × 10^9^/L) [12], requiring prophylactic platelet transfusion (in case of bone marrow suppression or in case of analytical interference) that cannot be overcome; however, it needs a flow cytometer and experienced technicians to be applied. The previous generation of hematology analyzers has counted on an immuno-platelet method using only one monoclonal antibody (CD61). Nonetheless, it has been reported equivalent to the reference method in most cases but Glanzmann Thrombasthenia (since in the latter case, the integrin αIIbβ3 is most often very reduced or even absent).

#### 2.1.3. Impedance Platelet Counting

Coulter principle, patented by Wallace Coulter in 1953, provided the basis for the first automated method of cell counting. During analysis, a constant direct current is established between two electrodes. When a blood cell, suspended in electrolytes solution, passes through a small aperture encompassed by the two electrodes, the impedance changes, causing a decrease of current intensity. This decrease will be recorded as a pulse. For a given sample volume, the number of recorded pulses corresponds to particle concentration. The amplitude of the pulse corresponds to the size of the individual particle or cell. This technique was first used for platelet counting in platelet-rich plasma rather than in whole blood since the coincidence with RBC was too high in the latter (RBCs interfered with platelets).

In the 1970s, the use of hydrodynamic focusing, a system that can overcome the recirculation problems in the sensing zone, and the use of pulse shape analysis enabled reliable count of platelets in whole blood samples [13].

Different brands of impedance-based analyzers are present on the market. They all enumerate the platelets’ number by counting the blood elements within a specific size range, but they differ in the latter. The analyzers generate a platelet volume histogram on which log normal curves are fitted, and final data are calculated (Figure 2). In this way, the analyzer extrapolates the platelet count in the area between 20 fL and 60 fL, avoiding interference with small RBC. However, inaccuracies of platelet count obtained from methods relying on cell size can result in overestimation or underestimation of the platelet count. A false increase of the platelet count can be observed in the presence of non-platelet particles (as small as platelets), such as fragmented erythrocytes, fragments of nucleated cells, bacteria, fungi, lipids, and cryoglobulins. Conversely, a false decrease in the platelet count can be observed in the presence of large platelets, exceeding the upper limit of the normal range of platelet size.

#### 2.1.4. Optical Platelet Counting

Optical light scatter techniques were applied in the eighties to count platelet with an automated hematology analyzer like ELT8^®^ or 800 from Ortho Instrument (Westwood) and Hemalog D^®^ from Technicon manufacturer (Technicon Instruments Corp., Tarrytown, NY, USA). In ELT 800^®^, the laser light scattering and hydrodynamic focusing improve platelet enumeration. The scattered light is directly related to the size (area), surface irregularities, and refractive index of the illuminated particle or cell. At a low angle of scattered-light detection, the area of the cell represents the highest input source to extrapolate cell volume. At high angle detection of scattered light, the refractive index is the major determinant. Based on this principle, the low angle scattered-light measurements provide better discrimination of particles that have the same volume but different contents, i.e., large platelets are discriminated from small RBC. When two detectors are used for low and high angle scattered-light measurements [14], the platelet size is represented by the low angle scattered-light measurement, whereas the platelet density is represented by the high angle measurements (Figure 3). The Advia^®^ instrument (Siemens) measures scattered light between 2–3° and 5–15° [15], the Cell-Dyn Sapphire^®^ one (Abbott Diagnostics Division, Santa Clara, CA, USA) measures at 7° and 90° [16]. The recent Allinity H^®^ from the same manufacturer performs measurements at multiple angles (seven light detectors) for better discrimination. In a study, including patients with hereditary macrothrombocytopenia, the optical-based counters were found more reliable and of higher clinical relevance than the impedance counters [2].

The discrepancies in mean platelet volumes (MPV) obtained by hematology analyzers could be partly explained by the differences between the measurement’s methods employed by the analyzers, i.e., impedance and light scattering. 

#### 2.1.5. Fluorescence Platelet Counting

This technique is employed by instruments from two manufacturers: Sysmex (Kobe, Japan), XE 2100^®^, XN^®^ instruments; Mindray (Shenzhen, China), BC-6800^®^ instruments. A fluorescent dye is used to stain young RBC (reticulocytes) and platelets (Figure 4). The simultaneous measurement of fluorescence and scattered light gives a more specific recognition and a more accurate count of platelets [17,18]. This is particularly true when large platelets have to be distinguished from other relatively large particles (small RBC, RBC fragments). However, there are still samples in which platelet count can be overestimated (see Section 3.4.1). In such cases, the immunological platelet count (reference method) remains the best way to count platelets. 

The fluorescence-based method measures also the fraction of young platelets [19]: so-called “reticulated” platelets with Cell-Dyn Sapphire^®^ and Allinity H^®^, or immature platelet fraction (IPF) with XE^®^ and XN^®^ Sysmex series and Mindray BC-6800^®^. The fluorescent dye is CD4K530 for Abbott analyzers and oxazine-based for Sysmex analyzers. Despite the poor standardization and a lack of correlation between the young platelet fraction measurements among the different instruments (absence of reference method and the existence of different reference ranges for different instruments), these parameters are good indicators of thrombopoietic activity [20,21,22]. IPF absolute count has been proven as an indicator of platelet recovery after chemotherapy in pediatrics [23] and, thus, has been proposed as a tool to categorize neonatal thrombocytopenia [24] or to predict peripheral immune thrombocytopenia [25,26]. IPF increases in diseases when there is increased platelet destruction or consumption and decreases in bone marrow failure. IPF is also correlated with the platelet size and represents an excellent alternative to impedance-derived MPV for detecting hereditary thrombocytopenias with large platelets [27,28]. The highest values of IPF in these hereditary diseases are not related to an increase of RNA content but to the staining of mitochondria and large granules [29]. From a therapeutic point of view, IPF has been shown to correlate with bleeding in patients with immune thrombocytopenia [30] and to predict response to thrombopoietic agents in these patients [31]. It has also been shown that immature platelets have a more reactive profile, thus making IPF a clinically interesting parameter in the prognosis of coronary artery disease [32,33]. 

### 2.2. Platelet Count Normal Reference Values and Thresholds for Checking and Verification

#### 2.2.1. Platelet Count Normal Reference Values

Platelet count normal reference value ranges from 150 to 400 × 10^9^/L as defined by the “Haute Autorité de Santé” (HAS) [34] (a French independent scientific public authority with legal personality) and international guidelines [35,36]. This range has recently been confirmed by Troussard et al. [37] in a study conducted with 33,258 adults. 

#### 2.2.2. Which Quantitative Thresholds should be Considered to Trigger Additional Actions before Reporting the Result?

In the case of thrombocytopenia (platelet count <150 × 10^9^/L), detected early or during the follow-up of a patient, the first thing to do is to look for interferences that might have falsely resulted in a low count (clumps, fibrin, macroplatelets, etc.), as discussed above. A significant difference in the platelet count, as compared with a previous count, should incite biologists to recheck. According to the curves published by Berend Houwen [38], delta-check is used when such difference exceeds 50% in adults and 30% in children. However, if thrombocytopenia is severe (<20 × 10^9^/L), the delta check is meaningless. If no interferences are found, a blood smear review is mandatory when an adult presents less than 100 × 10^9^ platelets/L in the initial investigation or >3 months after the initial result, and a child with less than 150 × 10^9^ platelets/L in the initial investigation or >1 month after the initial result [39,40]. 

In the presence of thrombocytosis (platelets >450 × 10^9^/L and absence of history) or a significant increase in platelet count without obvious cause (>50% compared with a recent result), the possibility of an analytical interference should be considered first. Such interference could come from the incapacity of some analyzers to discriminate between platelets and other particles of comparable size, or from an erroneous overestimation of platelets, probably generated by their density or diffraction. Smear review can be useful to detect interferences that could falsely increase the platelet count, as discussed later (cryoglobulins, fragments, etc.). If the recount gives the same results, the examination of the blood smear can point towards the diagnosis of a myeloproliferative neoplasm. However, checking the blood smear in this context is not mandatory since it will not provide formal evidence for the diagnosis [40]. If thrombocytosis has already been reported, a new blood smear review is not required.

#### 2.2.3. Pre-Analytical Errors

Spurious thrombocytopenia may erroneously be reported because of improper filling of the test tubes (too much or not sufficiently filled, making their inversion difficult) and/or insufficient inversion of the tube at the time of sampling. EDTA tubes require 8–10 inversions to thoroughly mix blood with the anticoagulant [41]. 

If sample coagulation is suspected, one should search for a clot in the tube. The tube content should be carefully examined and the inside of the cap and the walls of the tube as well. The content of the tube must be transferred into a hemolysis tube in order to look for any small clots. If a clot is detected in the tube, the whole complete blood count (CBC) must be rejected, signaling it as a “coagulated sample” and “pre-analytical non-compliance”.

## 3. Technical and Biological Validations of Platelet Count: GFHC Guidelines

### 3.1. Technical Validation: How to Check a Platelet Count?

There are several ways to check platelet counts when abnormalities are suspected.

Concerning the impedance method of platelet counting, if interference (pollution of the platelet distribution curve by fragmented RBC, very small RBC, cryoglobulin, cytoplasm fragments of nucleated cells, large platelets, microorganisms, lipids, etc.) is suspected [42], the count must be checked using another channel that the hematology analyzer is equipped with (optical or fluorimetric methods). Typical examples of interference are illustrated in Figure 5. If an alternative technique is not available, either automated microscope-based estimation or visual counting method using a counting chamber (Malassez, Thoma, Nageotte) must be considered, keeping in mind the laboriousness and relative inaccuracy of the latter. Ultimately, an immunological counting of platelets may be considered in case of persistent interference. If the technical interference is confirmed (discrepancy between the two techniques), a May–Grünwald Giemsa (MGG)-stained blood smear will be performed to document the interference.

If such abnormalities are associated with the reference anticoagulant EDTA (platelet clumps, leucoplatelet satellitism), as discussed above, another count might be ordered in a new blood sample with another anticoagulant in addition to a new EDTA tube count run in parallel. 

### 3.2. Biological Validation

Facing any abnormal results, the specialist in laboratory medicine [43] must ensure that all the necessary checks have been done and then interpret the result on its numerical value while integrating other data, such as past results, delta check, and clinical context. He must analyze all the available laboratory tests and integrate them into the clinical context. Later on, and based on the overall analysis of the medical and therapeutic records, the pathologist must provide his advice and, if necessary, directly contact the patient’s healthcare provider. 

### 3.3. Facing Thrombocytopenia or a Significant Decrease in Platelet Count

#### 3.3.1. Technical Validation: Search for Analytical Pitfalls that Cause Artefactual Reduction of the Platelet Count

The platelet count can be biased in a substantial number of cases. Platelet count can be reduced and remain within normal limits or may show false thrombocytopenia or artefactually accentuate true thrombocytopenia.

##### Platelet Clumps

● Definition, epidemiology, circumstances, and underlying mechanisms

EDTA-induced pseudothrombocytopenia is a rare in vitro phenomenon (0.07 to 0.20% of the general population and 0.1 to 2% of hospitalized patients) [44]. It is mainly caused by the presence of the patient’s anti-platelet antibodies, the effect of which is facilitated by EDTA, preferentially between 0 and 25 °C, i.e., in the EDTA tube. These antibodies are directed against platelet glycoproteins GPIIb/IIIa (against cryptic epitopes, which are revealed upon platelet contact with EDTA [45]) and cause platelet activation via tyrosine kinase, leading to clumps and, therefore, to false thrombocytopenia in vitro. However, this activation has no relationship with the increase in platelet activation involved in the pathogenesis of arterial thrombosis (myocardial infarction, or stroke), when platelet clumps are generated in vivo. The most important element is that EDTA-induced pseudothrombocytopenia is never accompanied by hemorrhagic signs. The presence of clumps varies over time for a given patient; periods with clumping alternate with periods without clumping with no obvious explanation.

Von Willebrand type IIB disease has been described as a very uncommon cause of in vivo clumping, which can also be evoked when platelet clumps are present in a blood smear drawn without any anticoagulant (e.g., finger prick) [46].

● When to look for platelet clumps?

The presence of platelet clumps in association with EDTA must be investigated in four situations [47,48]:

- The alarm of the analyzer is set off, indicating an anomaly concerning platelets (abnormal distribution of platelets, platelet clumps, abnormal platelet scattergram, etc.). However, the specificity and sensitivity of these alarms vary widely between suppliers, and careful analysis of the platelet distribution curve should not be overlooked. Regardless of platelet count, any abnormal distribution of the platelet curve should prompt a blood smear review and another method of counting if available. When platelet impedance histogram does not show a return to the baseline of at 20 fL, this is very suggestive of the presence of platelet clumps [49]. Depending on the size of the clumps, there may be no smoothing of the platelet curve (with Beckman–Coulter analyzer) or even a disturbance in the leukocyte count (spuriously raised).

- Positive past history of platelet clumps 

- A decrease in platelet count by more than 50% for an adult and ≥30% for a child when compared with previous results 

- In the case of thrombocytopenia (platelets <150 × 109/L) observed either in the initial investigation or appearing during the follow-up of the patient.

● How to search for platelet clumps?

The presence of platelet clumps can be checked either by examining a drop of fresh blood under a phase-contrast microscope or by using an MGG-stained blood smear (Figure 6a). A clump of platelets is defined by the presence of at least five attached platelets (GFHC consensus). The absence of platelet clumps should be explicitly mentioned in the laboratory report. 

The presence of rare clumps may indicate a sampling problem rather than EDTA-induced pseudothrombocytopenia. In the latter, the presence of large clumps in the fringes and edges are almost always detected.

Fibrin can generate the same alarms as platelet clumps do. Observing fibrin filaments in the smear suggests the presence of a microcoagulum. It is recommended to add a comment in the report to alert about probable underestimation of the platelet count and to suggest rechecking with a new sample.

● What to do in case of platelet clumps associated with EDTA-induced thrombocytopenia?


**How to report the result?**


It is helpless to count the platelets with another method on the same sample. In case of clumps, the result of the platelet count reported by the analyzer must be replaced by the note “clumps” with a comment requesting further check with EDTA and another anticoagulant (sodium citrate, CTAD (sodium Citrate, Theophylline, Adenosine, and Dipyridamole), magnesium sulfate, etc.) in parallel, depending on laboratory practice and available tubes. If necessary, the platelet count may be reported as a comment underneath “clumps”. If the platelet count is ≤20 × 10^9^/L, no number can be reported.

If the platelet count in a second EDTA blood sample no longer shows platelet clumps, this new EDTA-based count should be reported rather than the platelet count with another anticoagulant. Of course, as analyzers are calibrated with EDTA, only the platelet count can be reported using a different anticoagulant. If the clumps persist in EDTA blood sample, and the control tube with another anticoagulant does not show platelet clumps, the platelet count on the latter will be retained, specifying the anticoagulant used. In the case of clumps on both types of anticoagulants, no platelet count should be reported, and the relevance of a visual count using a capillary sample should be discussed with the clinician.


**Alternatives to EDTA**


Other anticoagulants can be considered: sodium citrate, CTAD, magnesium sulfate, sodium or lithium chloride unfractionated heparin (UFH) [44], sodium fluoride [47,49], ammonium oxalate [50,51], or CPT (trisodium Citrate, 5’-Phosphate pyridoxal, and Tris) [52,53]. Some of these anticoagulants need to be prepared extemporaneously. Another possibility is the addition of calcium chloride [44], anti-platelet drugs (acetylsalicylic acid, prostaglandin E_1_, apyrase, monoclonal antibodies directed against GpIIb/IIIa), potassium azide, kanamycin, amikacin, or other aminosides [54,55,56], which prevent the formation of clumps or dissociate them. 

Heparin (UFH) is not a recommended alternative to EDTA because it makes blood smear examination impossible due to cellular halos formation. Heparin can also cause/promote platelet activation, and hence cause spurious thrombocytopenia [57].

The most commonly used alternative anticoagulant in the case of EDTA-induced thrombocytopenia is sodium citrate, thanks to its wide commercial availability, unlike the others. However, sodium citrate is in liquid form, which introduces a dilution factor for the platelet count. The citrate tube must not be centrifuged beforehand. Moreover, the platelet count with a citrate tube is not stable over time and is often underestimated [51] in samples analyzed more than three hours after sampling. For some patients (15 to 20%), platelet clumps may even be present in the presence of sodium citrate [50].

The usual practice is to correct the count of platelets with regard to the dilution factor introduced by the anticoagulant (one volume for nine volumes of blood). However, a study showed that this correction factor was not accurate and underestimated platelet count compared to EDTA [51]. Another study showed that the correction factor for overcoming this underestimation was 17% within three hours after sampling [58]. However, given the lack of external quality control, as well as a reference sample for platelet count with a citrate tube, it would be desirable that each laboratory conducts a correlation study between platelet count in EDTA and citrate blood to estimate its own commutability factor. In practice, this study is difficult to conduct, and the GFHC co-opts the current 10% factor and recommends caution if the sample is analyzed more than three hours after sampling. The CTAD mixture is another commonly used EDTA-alternative, and several studies have shown that this anticoagulant is a good solution when dealing with platelet clumping with EDTA [59,60,61]. The long-term stability of the platelet count is better, but the problem of dilution persists, not mentioning the increased cost.

An interesting alternative is to perform platelet count using magnesium sulfate. This anticoagulant, which was historically the first to be used for platelet counting (before EDTA), is an excellent alternative in case of platelet clumps formation with EDTA alone or with other anticoagulants (multiple intolerances) and has the advantage of ensuring the stability of the platelet count for at least 12 h [62,63].

If platelet clumps arise with EDTA and other anticoagulants, the alternative is using a native whole blood sample (finger prick without any anticoagulant) for a manual count. The use of special mixtures containing various molecules (dextrose, theophylline, aminoglycosides) has been proposed but remains difficult to implement due to the laborious aspect of the method. Overall, the faster the EDTA tube is analyzed, the lower the risk of clumps formation will be. Besides, maintaining the sample at 37 °C after being drawn and during analysis [64] can sometimes also prevent clump formation. On the other hand, incubating a tube containing clumps at 37 °C will not make them disappear. 

##### Large Platelets. Definition, Circumstances, and How They Disturb the Platelet Count

The size of a platelet is assessed by comparison with an RBC or by using a micrometer. Accordingly, a large platelet (macroplatelet) is a platelet, the size of which is between a half and an entire RBC (4–8 µm). A giant platelet is as big as or larger than an RBC (≥8 µm) (Figure 6b) [65,66,67]. Macroplatelets or even giant platelets are found in various conditions, whether or not thrombocytopenia exists (myeloproliferative neoplasms, myelodysplastic syndromes, inherited platelet disorders and/or thrombocytopenias, immune thrombocytopenia), and often lead to underestimation of the platelet count. Their detection is crucial to help to correct the platelet count and to guide the diagnosis in case of constitutional thrombocytopenia [68]. 

● Benefits and limitations of the mean platelet volume (MPV)

Thanks to automated analyzers, MPV has been widely used in clinical practice despite its high sensitivity to pre-analytical variables, like the type of anticoagulant, the analysis time, and the pre-analysis temperature of storage. MPV depends largely on the platelet counting technique (impedance or optical) used in the automated analyzer and on the conditions of sample dilution before counting: osmolality, temperature, type of detergent, which explains the high discrepancy between counters [69]. 

Calibration of automated analyzers for MPV is independent of their calibration for RBC volume (MCV) and is most often established with a suspension of various size latex beads. For all these reasons, reported MPV in healthy people highly varies in the literature: from 6.0 to 13.2 fL [70,71,72].

Nevertheless, MPV remains an interesting clinical parameter to guide blood smear examination. In 1982, David Bessman showed a correlation between MPV and megakaryocyte ploidy [73]. In patients with immune thrombocytopenia (low platelet count, MPV above the normal limit, and high megakaryocyte ploidy) and those with reactive thrombocytosis (high platelet count, low MPV, and low megakaryocyte ploidy), the relationship between MPV and the platelet count has resembled or exceeded the relationships found in normal subjects [74]. By contrast, in patients with thrombocytopenia resulting from bone marrow failure, MPV and megakaryocyte ploidy are substantially lower than in normal people or the above-mentioned patients. This result was confirmed in well-defined conditions (impedance-based count on Sysmex XE 5000), by Till Ittermann et al. in 2019 who stated that since MPV inversely correlates with the platelet count in normal subjects [75], its value must be interpreted in relation to the platelet count.

Finally, it is important to carefully examine the histogram of the platelet population distribution. A study showed that the absence of smoothing of the platelet histogram was the most sensitive anomaly, indicating the presence of macroplatelets [48]. In some (very rare) cases, a platelet can reach the size of a leukocyte.

● How to report the result?

Giant platelets (≥8 µm or the size of an RBC) can give an underestimation of the platelet count, especially on impedance-based hematology analyzers. It is imperative to perform an MGG-stained smear and, if possible, a platelet count by another method (fluorimetric methods, manual direct method, or automated microscope). Macrocytic and giant platelets must be reported on the CBC report, and a platelet formula count can be detailed (macro and giant platelets are counted in at least 100 platelets).

##### Platelet Satellitism

● Definition, circumstances, and how they disrupt the platelet count

Satellitism is a rare (1 in 12,000 blood samples) in vitro acquired phenomenon, typically seen in EDTA anticoagulated blood, and induced by adherence of platelets to mature neutrophils, forming rosettes (Figure 6c) [42]. Platelet satellitism has also been reported around lymphocytes or around lymphoma cells [76,77]. More rarely, platelet satellitism is reported specifically around other cells, e.g., around monocytes in blood samples anticoagulated with EDTA or/and heparin [78,79], around basophils in chronic myeloid leukemia [80], or around eosinophils [81]. Other EDTA-related situations have been described as satellitism, but actually, they represent either cluster of lymphocytes or RBC surrounding lymphocytes or neutrophils in the presence of EDTA [82,83]. It is not consistently linked to a given clinical entity but can be the cause of spurious thrombocytopenia. It is sometimes associated with an autoimmune process mediated by IgG, with or without the involvement of gamma Fc receptors of polymorphonuclear neutrophils (PMN) [84,85]. The involvement of the platelet membrane alphaIIb/beta3 complex (GPIIb/IIIa) has also been noticed [84], as has been the presence of IgG autoantibodies that are directed against a cryptic antigen found on the platelet alphaIIb/beta3 complex and on the neutrophil Fc gamma III receptor (CD16), and possibly revealed by the presence of EDTA [84]. The presence of cryofibrinogen and the involvement of thrombospondin [80,85] could also be encountered in this phenomenon. As such, probably, platelets adhere to the PMN membrane and then detach over time as platelet clumps [42,86].

There is no specific warning message: in most cases, it is the biparametric histogram of the leukocyte differential that shows an abnormal location of PMN cells, often of unusually large size, and the warning message indicates the abnormal location of PMN cells and/or the presence of immature granulocytes (abnormal large neutrophils). 

● How to report the result?

So far, satellitism has not been reported with citrate anticoagulant, the use of which might be recommended, among others. The underestimation of the platelet count is mild and can be neglected, although cautious examination of blood smear for platelet clumps must be performed. In any case, satellitism should be mentioned in the report, describing the leukocytes involved. A practical attitude when facing thrombocytopenia is proposed in a decision tree (Figure 7). 

#### 3.3.2. Facing Real Thrombocytopenia: Which Investigations Are Necessary? How Can We Guide Clinical Management?

Two parameters have to be considered first: the severity of thrombocytopenia and if other abnormalities of the CBC and differential are present. A platelet count of less than 20 × 10^9^/L is associated with high bleeding tendency and must be urgently discussed with the physician. Thrombocytopenia can be caused by four mechanisms: hemodilution, hypersplenism, bone marrow failure, or reduced platelet lifespan. Hypersplenism can easily be suspected in patients with mild to moderate thrombocytopenia associated with neutropenia, anemia, and splenomegaly. In this context, as in others, comprehensive and accurate clinical information (medication, history of hepatic or renal disease, splenomegaly, age at onset of thrombocytopenia, family history, etc.) is crucial. Whether thrombocytopenia is the only sign or not, careful examination of the blood smear is mandatory and, as already mentioned, both the International Society for Laboratory Hematology and the GFHC recommend smear review when the platelet count is less than 100 × 10^9^/L in adults and <150 × 10^9^/L in children, among other criteria [39,40]. We strongly recommend analyzing the morphology of WBC and RBC in addition to that of platelets as all lineages can be informative for the diagnosis even in isolated thrombocytopenia. Briefly browsing the wide range of differential diagnoses, WBC examination can reveal blast cells, abnormal lymphoid cells, septic modifications, dysplastic features, Döhle body-like inclusions; RBC examination can reveal reticulocytes (polychromasia in the context of regenerative anemia), schistocytes, dacryocytes, plasmodium, babesia; platelets can show abnormal volume (increased or decreased) or abnormal granule content. Typical examples are illustrated in Figure 8. When thrombocytopenia is not alone or when the blood smear is not in favor of a specific diagnosis and depending on the severity of thrombocytopenia and the patient’s presentation (age, splenomegaly, medications, alcohol intake, immunodeficiency syndrome, autoimmune disease, etc.), bone marrow examination can be ordered. Various conditions can be diagnosed at this step, including malignant and non-malignant hematological or non-hematological diseases. Immune thrombocytopenia (ITP) is defined as a platelet count below 100 × 10^9^/L in a patient for whom other causes of thrombocytopenia have been ruled out [87]. An important point to highlight is that the incidence rate of ITP is 2–4 cases per 100,000 person-years [87], roughly ten times higher than inherited thrombocytopenia (IT), which has been estimated at 2.7 cases per 100,000 individuals in the Italian population [88]. The introduction of high throughput sequencing techniques (formerly known as next-generation sequencing (NGS)) has greatly broadened knowledge on IT over the past few years, during which more than thirty different IT forms have been identified [89]. We strongly recommend considering IT whenever the acquired origin of thrombocytopenia is not obvious. Cases with severe, mild to moderate thrombocytopenia can be misdiagnosed as ITP, and a recent cohort on 181 women with IT has shown that 31% of them were misdiagnosed as ITP and received undue therapies [90]. For IT investigation, evaluating the platelet size on peripheral blood (PB) smears can be helpful. Although a small percentage of large platelets is a common finding in ITP, a significant proportion of giant platelets (>5%) should strongly orient the diagnostics towards IT, considering at first MYH9-RD, since it is the most prevalent IT worldwide (see [89] for a review on IT), and Bernard–Soulier syndrome (biallelic form). Moreover, bleeding is not the only clinical complication in patients with IT as several IT forms (FDP/AML, ANKRD26-RT, and ETV6-RT) are predisposing conditions to blood malignancies; MYH9-RD predisposes to end-stage renal disease, deafness, and presenile cataract, and other ITs predispose to bone marrow failure. For all these reasons, documenting the acquired (or non-acquired) origin of thrombocytopenia is critical in diagnostics. 

### 3.4. Facing Thrombocytosis or a Significant Increase in the Platelet Count Compared with a Previous Result

Thrombocytosis is a common finding in a medical laboratory. As normal platelet count values are between 150 and 400 × 10^9^/L, thrombocytosis is often considered for platelet counts >450 × 10^9^/L on two successive blood counts [91]. The diagnostic approach focuses on eliminating a reactive etiology, by far the most common, before considering the existence of primary thrombocytosis. However, in the absence of a history of thrombocytosis or a significant increase in the platelet count without obvious cause (>50% compared with a recent result), the possibility of analytical interference should be raised, as discussed before. The inability of hematology analyzers to discriminate between platelets and other particles of comparable size, density, or diffraction may lead to an erroneous overestimation of platelets.

#### 3.4.1. Technical Validation Step: Research and Identification of Interferences that could Result in Overestimation of Platelets

Several types of interferences can induce false thrombocytosis, or inversely incur artificial total or partial masking of thrombocytopenia [42]. In practice, their occurrence is suspected by an abnormally high and unexplained platelet count or unexpected increase. Taking into account the alarm messages of analyzers, the interpretation of mono- and bi-parametric histograms associated with platelet and erythrocyte counts and the inter-comparison of possible additional counting channels of platelets are decisive elements to check interference. Blood smear examination may provide guidance as to the nature of the interference. Although these interferences are rare or even very rare (depending on patients’ recruitment), they can lead to spurious platelet count.

##### Cryoglobulins

Cryoglobulins are usually immunoglobulins or immunoglobulin complexes, sometimes mixed with complement fractions, characterized by precipitation or gelation at a temperature below 37 °C and dissolution after reheating to 37 °C. Their presence is usually associated with lymphoproliferative syndromes (especially Waldenström macroglobulinemia), autoimmune diseases, hepatitis C, or any disease with circulating immune complexes.

Cryoprecipitates disturb the reading of the blood count, and their interference is size-dependent since small precipitates can induce pseudo-thrombocytosis or mask thrombocytopenia. Detecting cryoglobulin-related interference is not easy and quite often delayed, e.g., count results might be normal (or falsely normal), but the analyzers do not always set off the alarm. Such interference is neither constant nor related to cryoglobulin level or its nature and depends greatly on the type of technology implemented on the analyzer. Disturbances are often more pronounced on analyzers using reagents at laboratory temperature, although it could happen on all types of instruments.

It is recommended to look for cryoglobulinemia in any new case of thrombocytosis or unexplained changes in the platelet count over several consecutive blood counts. A surplus of small particles on mono- or bi-parametric platelet distribution histogram is sometimes observable and can be suggestive (Figure 5a). On analyzers equipped with two platelet counting channels, of which one is thermostatically controlled between 37 and 41 °C, a discrepancy in the numbers of platelets provided by each method constitutes a useful alert to search for cryoglobulins.

Microscopic examination is recommended if interference is suspected. Cryoprecipitates are, in almost all cases, visible when fresh blood is examined under a phase-contrast microscope (the anomaly is more obvious if the analysis is performed at low temperature), yet are barely visible on MGG-stained smears. Various morphological aspects are reported: clusters of dense and amorphous particles or puddles and the appearance of crystals or more or less pink globules. Most often, cryoprecipitates are translucent and colorless but easily detectable by a particular morphological defect they form on RBC (“pitted surface of RBC”, see Figure 9a).

In practice, the interference disappears when the sample is heated at 37 °C for at least 30 min, followed by rapid reanalysis. The anomaly is amplified in an aliquot of blood incubated at 4 °C, which confirms the presence of cryoglobulin. In some cases, cryoprecipitates persist after incubation at 37 °C, or the large ones are partially dissolved by the heat, generating an army of small elements that have high interference with platelets. In such situations, it is better to draw a new sample, maintain it immediately at 37 °C, and analyze it promptly.

Once the anomaly is detected and corrected, it is advisable to keep a trace of it in the patient record to help anticipate corrective actions for future analyses of subsequent samples.

##### Extreme Microcytosis and Red Cell Fragments 

The platelet and red cell counts are performed on the same channel(s) of the hematology analyzers. Under normal conditions, the particle size or refractive index differs significantly, allowing an easy distinction between platelets and RBC. In contrast, in pathological situations associated with the presence of numerous very small RBC (severe microcytic iron deficiency anemia, microangiopathic hemolysis with numerous schistocytes, or microspherocytosis due to extensive acute burns), some of these particles can be wrongly classified as platelets [92]. RBC fragmentation can also be seen in inherited RBC membrane disorders like pyropoikilocytosis, as illustrated in Figure 9b.

The alarms generated by the analyzers in these circumstances reflect the device’s inability to properly distinguish platelets from RBC (e.g., suspicion of giant platelets or platelet clusters, abnormal platelet distribution curve). Examination of the histograms of platelet and erythrocyte volumes is crucial. Situations, involving i) no return to the baseline of the platelet volumetric histogram beyond 20 fL, ii) or the detection of a population of microcytes on the platelet histogram, iii) or an excessively high value of MPV, iv) or very wide distribution of erythrocyte volumes, require blood smear examination and microscopic counting and/or the use of another counting method. Better separation between platelets and microcytic RBC or erythrocyte fragments is achieved in most cases using techniques based on optical light scatter, fluorescence, or flow cytometry [93].

##### Cytoplasmic Debris from Nucleated Cells 

These are almost always small fragments generated by malignant cells, either blastic (monoblast, lymphoblast) or lymphomatous cells (often during the leukemia phase of diffuse large cell lymphoma, sometimes during hairy cell leukemia) [79,94,95]. Similar to RBC fragments, when present in large numbers, they can sometimes artificially increase the platelet count. In our experience, monoblastic leukemia and Diffuse Large B-Cell Lymphoma (DLBCL) are the most common causes of this rare interference. The consequence is rarely thrombocytosis, but, much more often, they mask or partially correct the platelet count, which can delay platelet transfusion. Such a hypothesis should be brought up by both the clinician and the specialist in laboratory medicine when there is bleeding, yet the platelet count has not collapsed.

The analyzer result is usually coupled with non-specified warning messages (e.g., suspicion of large platelets or platelet clumps) or various other messages concerning the device’s inability to produce a fitted curve. The platelet count is distorted in both impedance and optical diffraction techniques. Measurement with fluorescent labeling of platelets is more accurate, but platelet specific immuno-counting may be necessary in some cases. Cytoplasmic fragments are easily detectable on MGG-stained blood smear, where they appear more heterogeneous in size and content than platelets and often more basophilic (see Figure 9c); a result close to the actual platelet count can be obtained by counting the number of fragments present per 100 platelets on the blood smear and then correcting the raw platelet count provided by the analyzer.

##### Lipids

Lipid micelles might be detected in case of postprandial sampling or intravenous parenteral nutrition and may sometimes interfere with the platelet count. The false increase in the platelet count is generally small and negligible in patients with normal platelet counts. However, it may have a greater impact on thrombocytopenic patients. Checking the alert messages from the analyzers and a careful analysis of the histograms are crucial to assess the potential impact of lipids on the platelet count. The methods proposed to specifically remove lipids from the sample should be avoided as they may themselves incur similar errors in the results, both by default and by excess. Alternative counting methods (blood smear, specific immuno-counting) could be useful in case of severe thrombocytopenia.

##### Microorganisms

Although very rare, false increases in the platelet count are reported in samples containing bacteria or yeasts [96]. It is usually the blood smear examination that highlights the anomaly and alerts to an error in the number of platelets. Such interference may show itself as an excess of small particles (bacteria or clusters of bacteria) on the platelet volume histogram. The results obtained by analyzers using a minimum threshold for defining platelets set at 3 fL or mobile, 2 to 6 fL, are less affected by this type of interference.

Large numbers of germs can either be present in vivo in patients with severe sepsis or result from their multiplication in the tube before it passes through the analyzer. The use of a non-sterile sampling tube in which microbial growth has occurred may also be considered. In addition, contamination of reagents by microalgae can similarly distort counting. Cleaning procedures and background noise measurement techniques recommended by manufacturers generally limit these incidents.

##### How to Report the Result?

Any identified interference that makes it difficult to determine the platelet count should be clearly stated on the blood count report with the name of the technique used to perform that platelet count. 

#### 3.4.2. Facing Real Thrombocytosis: Which Investigations are Necessary? How can We Guide Clinical Management?

Clinically, thrombocytosis is classified as "mild" for platelet counts between 450 and 700 × 10^9^/L, "moderate" between 700 and 900 × 10^9^/L, and "severe" or "extreme" for counts greater than 900 and 1000 × 10^9^/L, respectively [97]. Given the potential thrombo-hemorrhagic complications induced by the latter, it is recommended to quickly transmit such results to the prescribing physician so as not to ignore a potentially risky situation.

The diagnostic procedure is summarized in the decision tree (Figure 10), guided initially by the collection of patient-specific information (past history, anamnesis, and clinical and therapeutic data) and by some simple laboratory investigations if necessary [98,99]. As already mentioned, blood smear examination is not mandatory (if the platelet count is considered consistent) since it does not provide strong evidence for the diagnosis of myeloproliferative neoplasm, especially if thrombocytosis is the only presenting sign [40]. If thrombocytosis is not the only sign, blood smear can show abnormalities that may suggest infection (hypergranular neutrophils, myelemia), iron deficiency anemia (microcytosis, hypochromia, poikilocytosis), hyposplenism (Howell–Jolly body), or clonal hemopathy (large platelets, megakaryocytic fragments, erythromyelemia and dacryocytes, blasts, granulocytic dystrophies, excess of basophils, etc.) [100].

Keeping in mind that around 90% of thrombocytosis cases are reactive, in the absence of obvious clinical context (surgery, hemorrhage, splenectomy, medications, etc.), as reviewed by Harrison CN et al. [98], laboratory investigations should be performed looking for iron deficiency and/or inflammation. If thrombocytosis persists with no evidence of iron deficiency or inflammation, complementary investigations are to be done looking for myeloproliferative neoplasms. For instance, detecting *BCR-ABL1* transcript, the causative agent of rare chronic myeloid leukemia mimicking essential thrombocytemia (ET), and *JAK2*, *CALR*, *MPL* mutations, which are present in three myeloproliferative neoplasms: ET, primary myelofibrosis, and polycythemia vera [101,102,103]. Bone marrow biopsy examination, included in the latest WHO criteria [101], can be used to distinguish pre-fibrotic primary myelofibrosis from ET and polycythemia vera since each has its typical biological and clinical presentation. Other myeloid neoplasms that can show thrombocytosis are myelodyplastic syndrome, associated with 5q- or 3q abnormalities, and myelodysplastic/myeloproliferative neoplasm like sideroblastic anemia with thrombocytosis or chronic myelomonocytic leukemia. Rarely, thrombocytosis is primitive, non-clonal, and mostly with positive family history [104]. Extensive genetic testing is indicated in this context to detect germline mutations of *THPO*, *JAK2,* or *MPL* [105].

Another challenge regarding thrombocytosis is the evaluation of hemorrhagic or thrombotic risk. The platelet count is not a good marker to evaluate the risk of thrombosis in myeloproliferative neoplasm. Reactive thrombocytosis is usually not associated with hemorrhagic or thrombotic risks, but when the platelet count is >1000 × 10^9^/L, low-dose aspirin or other anti-platelet drugs can be discussed according to the clinical context. ET is associated with a high risk of thrombosis, and, for such, ELN (European Leukemia Net) recommends [106] that the therapeutic approach can be reoriented from observation alone, low-dose aspirin, to cytoreductive treatment plus low-dose aspirin. High counts of platelets are associated with hemorrhagic risk in the context of acquired Von Willebrand disease (AVWD). In ET patients, with platelets higher than 1000 × 10^9^/L and/or if the clinical manifestation includes bleeding, von Willebrand factor antigen level and ristocetin activity should be assessed. If the diagnosis of AVWD is confirmed, the use of low-dose aspirin is contraindicated [107]. 

## 4. Unanswered Questions and Concluding Remarks

Some questions remain open: what is the best anticoagulant in the case of EDTA-induced pseudothrombocytopenia? Which alternative method should be preferred in case of analytical interference? With the growing development of artificial intelligence, what will be the place of digital microscopes in our labs in the future?

The above recommendations on the platelet count are derived from the reflection of a working group on behalf of the GFHC group and supported by as a "strong professional agreement”. We wanted, first, to focus on the pre-analytical and analytical pitfalls encountered during platelet counting and are well known by laboratory biologists, then to propose a standardized management protocol for each. However, and given the diverse types of automated analyzers available at hand, such protocols cannot be rigid. Finally, we proposed decision-making trees for thrombocytopenia and thrombocytosis in order to homogenize practices and, above all, to achieve the most accurate reporting of the platelet count in these situations. These decision-making trees provide general bases that could be adjusted to fit the local practices of each laboratory and to take into account the technology used in the analyzers, as well as the most commonly used anticoagulants. It is up to everyone to "customize" one’s work in compliance with the general philosophy.

## Figures and Tables

**Figure 1 jcm-09-00808-f001:**
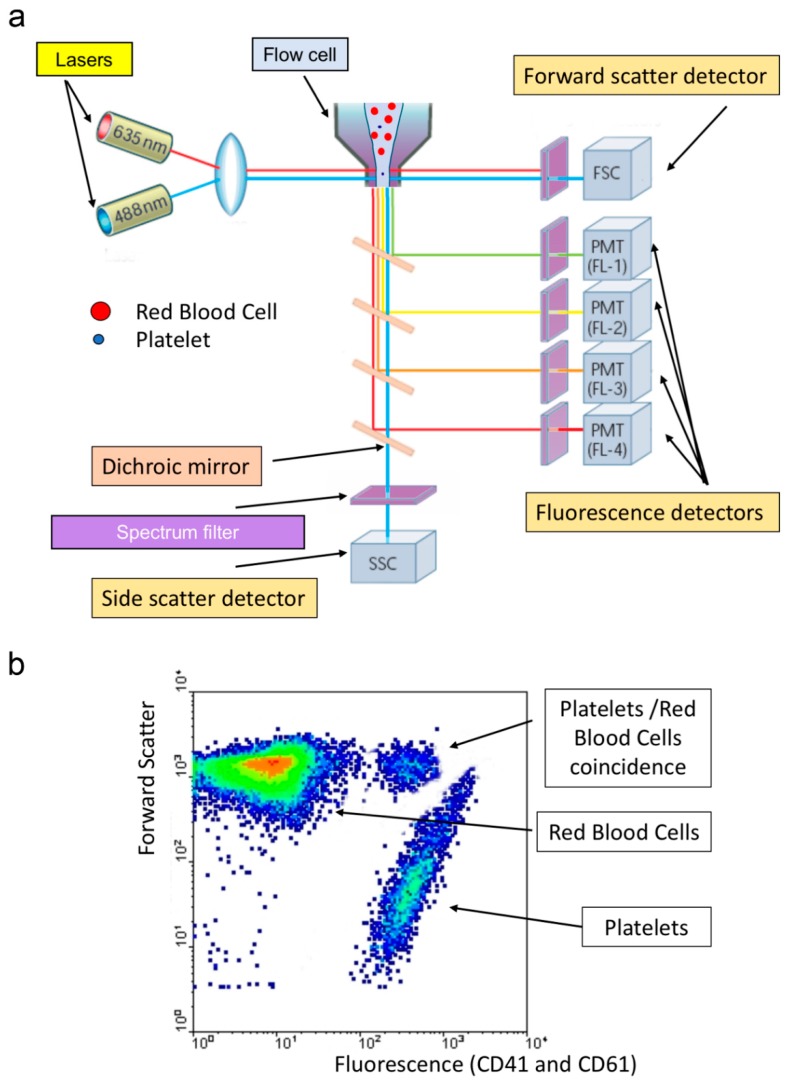
Immunoplatelet counting. (**a**) Illustration of the main optical elements of a flow cytometer. Cells in suspension flow in a single-file are illuminated in the flow cell where they scatter light (forward scatter: signal depends mainly on cell size, and side scatter: signal depends on complexity and granularity) and emit fluorescence. (**b**) Typical biparametric dot plot showing platelets, identified by their low forward scatter, compared to red cells, and their staining by CD41 and CD61.

**Figure 2 jcm-09-00808-f002:**
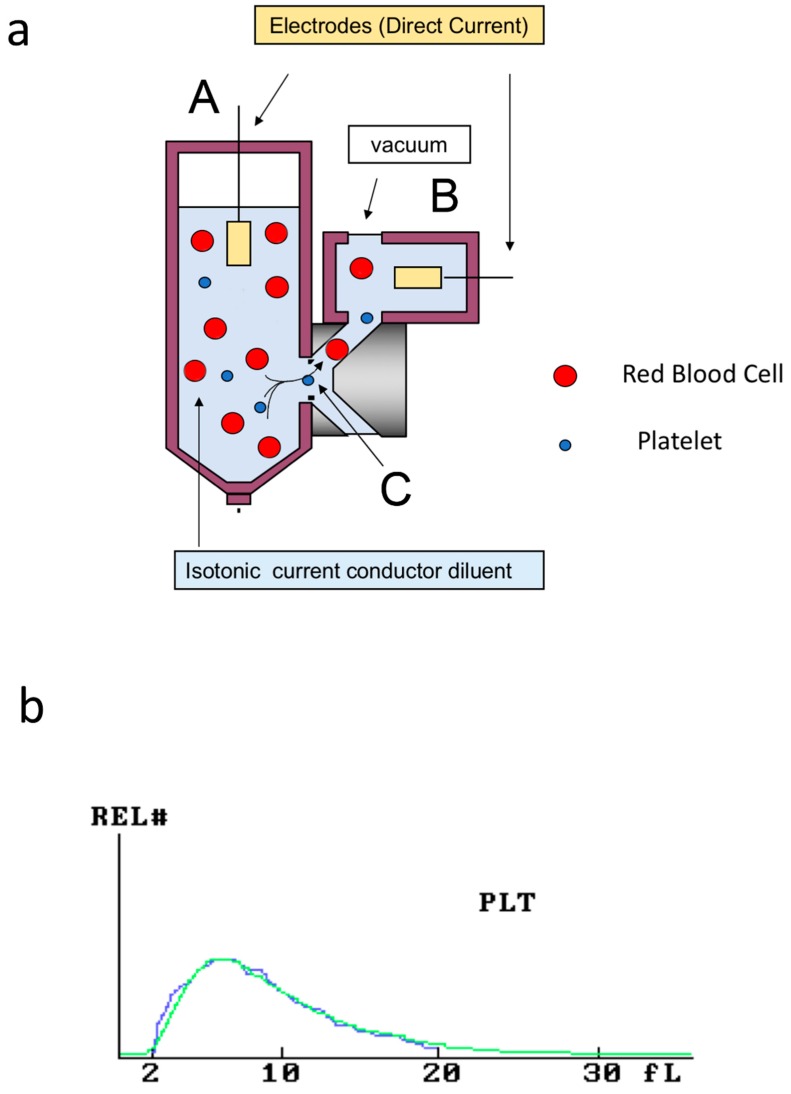
Impedance platelet counting. (**a**) Illustration of the Coulter principle: when a blood cell in suspension in an electrolytes-containing solution passes from part A to part B of the device through a small aperture (C) encompassed by two electrodes, the impedance changes, causing a decrease of current intensity. This variation is proportional to the cell volume and enables the enumeration of different blood cell types. (**b**) Typical monoparametric platelet histogram showing raw data in blue and derived log-normal curve in green.

**Figure 3 jcm-09-00808-f003:**
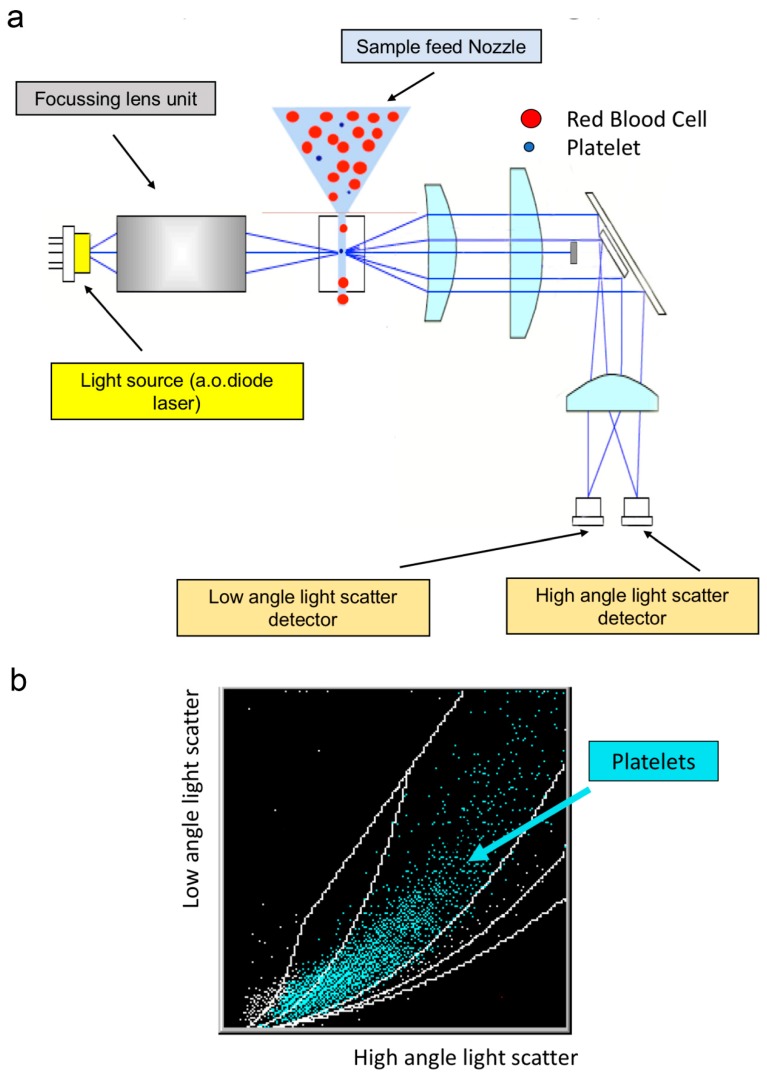
Optical platelet counting. (**a**) Illustration of the main optical elements. Cells in suspension flow in a single-file, and when illuminated in the flow cell, they scatter light (forward scatter: the signal depends mainly on cell size, and side scatter: the signal depends on the complexity and granularity). (**b**) Typical biparametric dot plot showing platelets in turquoise and debris in white.

**Figure 4 jcm-09-00808-f004:**
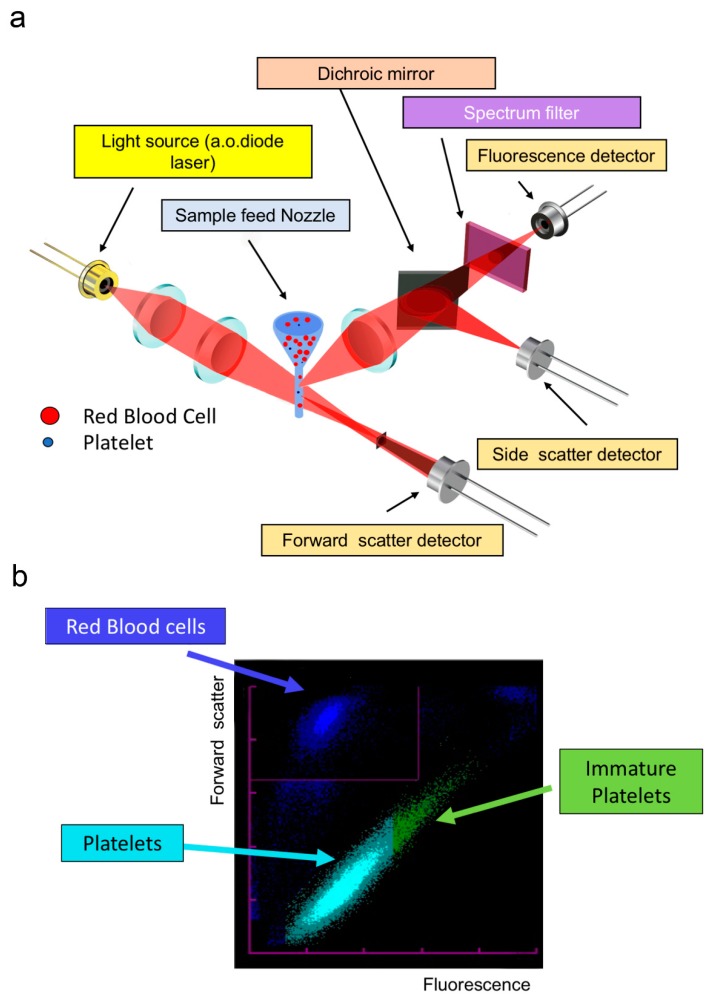
Fluorescence platelet counting. (**a**) Illustration of the main optical elements. Cells in suspension flow in a single-file, and when illuminated in the flow cell where they scatter light (forward scatter: the signal depends mainly on cell size, and side scatter: the signal depends on the complexity and granularity) and fluorescence. (**b**) Typical biparametric dot plot showing platelets in turquoise, immature platelets in green, and RBCs in blue.

**Figure 5 jcm-09-00808-f005:**
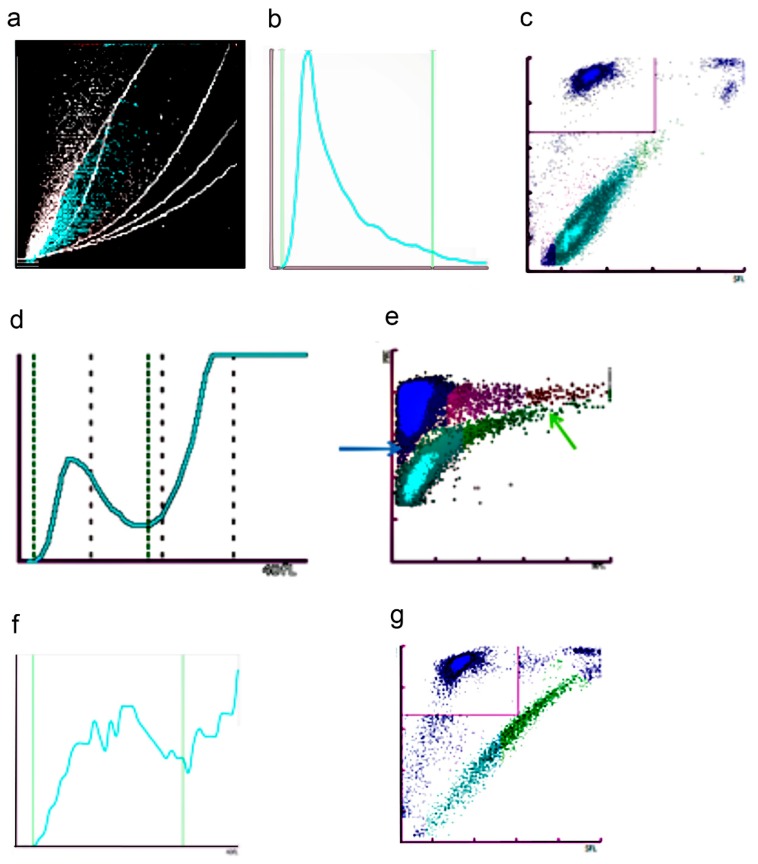
Typical examples of spurious platelet counts. Cryoglobulin: (**a**) spurious optical count (Advia 2120i^®^) 682 × 10^9^/L, (**b**) spurious impedance count (Sysmex-XN^®^) 735 × 10^9^/L, (**c**) correct fluorescence count (Sysmex-XN^®^) 171 × 10^9^/L. Small RBC in iron-deficient anemia, (**d**) spurious impedance count (Sysmex-XN^®^) 536 × 10^9^/L, absence of a return to baseline, and incapacity of the analyzer to clearly position a cut-off point (green dashed lines) for elements that may correspond either to large platelets (or platelet clumps) or to microcytic RBC or fragments of RBC, (**e**) In optical mode, the number of platelets is lower (432 × 10^9^/L), and the graph shows the better separation between small RBC (blue arrow) and large platelets (green arrow). Giant platelets: impedance count is underestimated (**f**) 23 × 10^9^/L, while the giant platelets are better enumerated with the fluorescence count (**g**) 32 × 10^9^/L.

**Figure 6 jcm-09-00808-f006:**
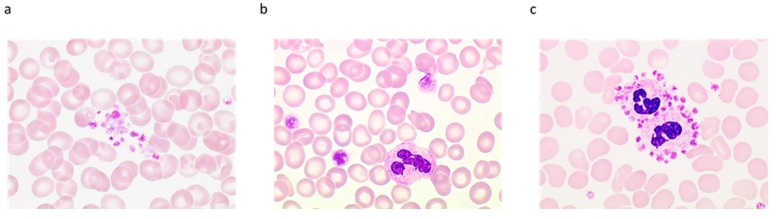
Examples of underestimation of the platelet count. (**a**) platelet clumps, (**b**) macro and giant platelets, (**c**) platelet satellitism.

**Figure 7 jcm-09-00808-f007:**
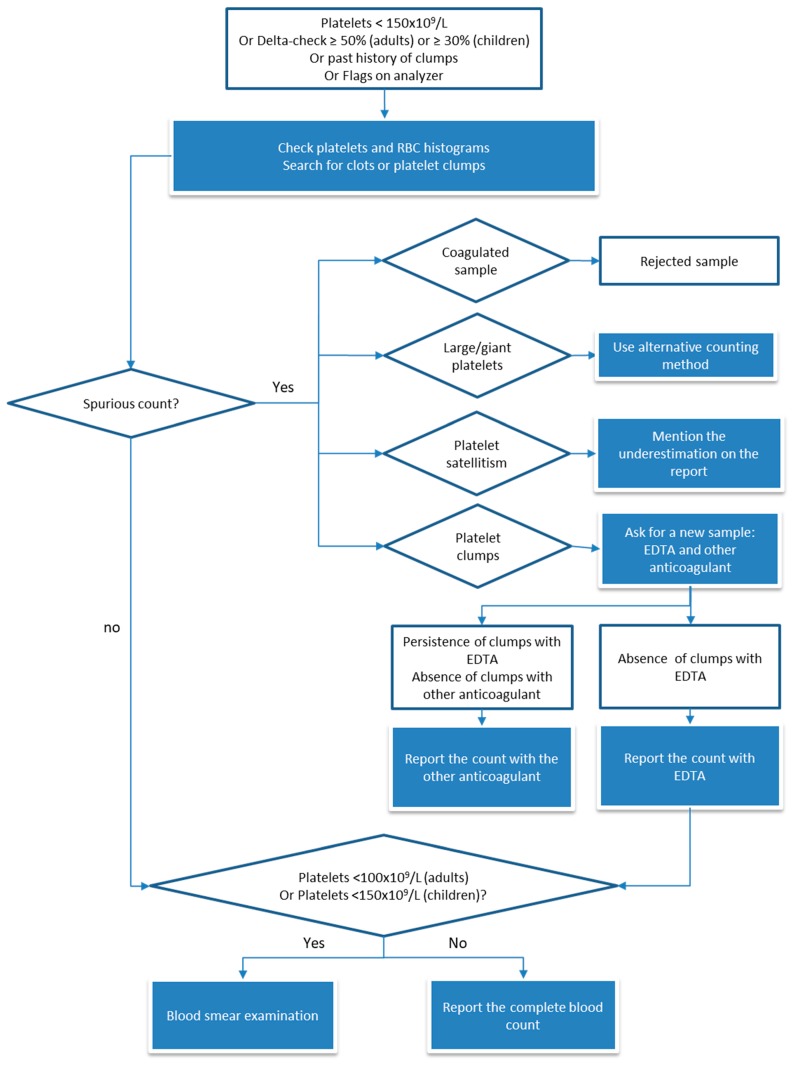
Decision tree for thrombocytopenia.

**Figure 8 jcm-09-00808-f008:**
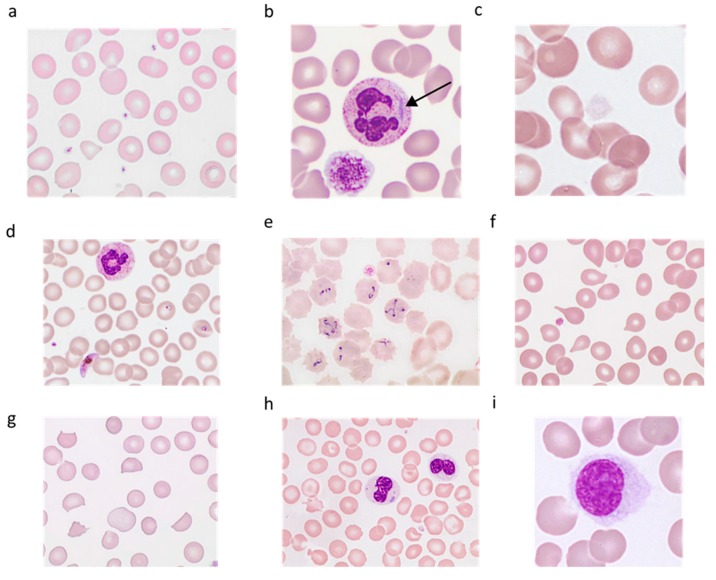
Illustration of several diseases revealed by thrombocytopenia (isolated or associated with other cytopenias). (**a**) X-linked thrombocytopenia with microplatelets (Wiskott–Aldrich syndrome-related disorder), (**b**) MYH9-RD with giant platelets; a Döhle body-like inclusion is indicated by an arrow, (**c**) Gray platelet syndrome with platelets lacking alpha granules, (**d**) *Plasmodium falciparum* infection, (**e**) *Babesia microti* infection (courtesy of www.hematocell.fr), (**f**) Dacryocytes, (**g**) Schistocytes, (**h**) Myelodysplastic syndrome with multilineage dysplasia, (**i**) Hairy cell leukemia.

**Figure 9 jcm-09-00808-f009:**
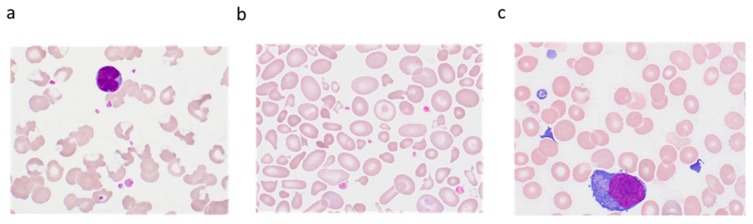
Interferences, incurring overestimation of the platelet count. (**a**) Cryoglobulins, (**b**) Red cell fragments in a case of hereditary pyropoïkilocytosis, (**c**) Cytoplasmic fragments of white blood cells in a case of diffuse large B cell lymphoma.

**Figure 10 jcm-09-00808-f010:**
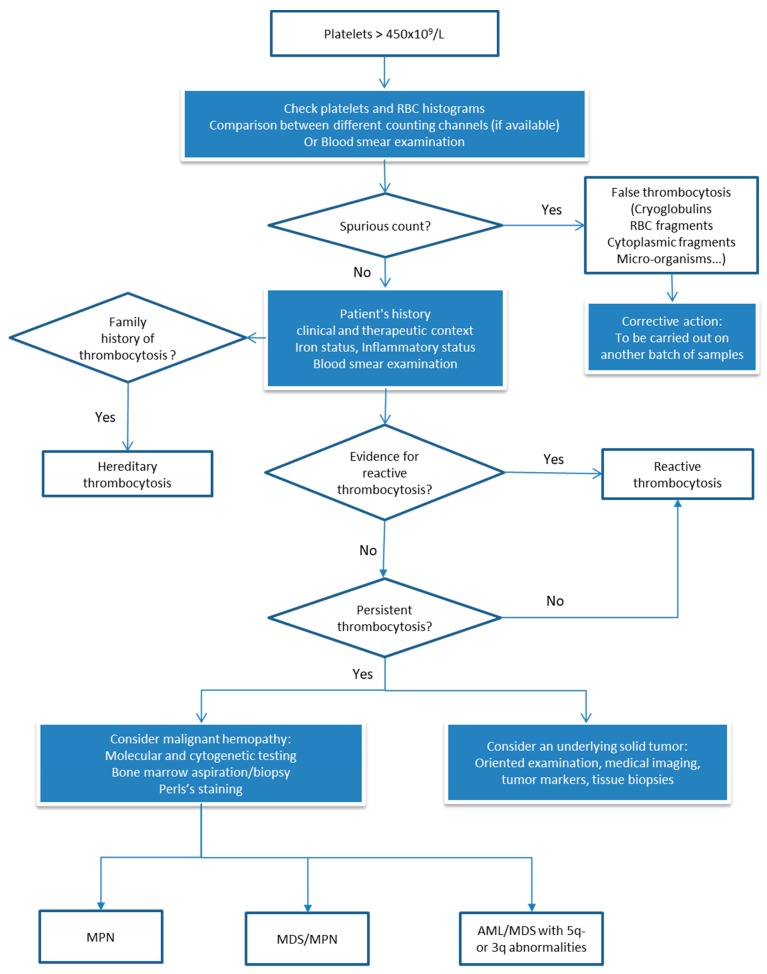
Decision tree of thrombocytosis.

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
