# Peer review of "Platelet Counting: Ugly Traps and Good Advice. Proposals from the French-Speaking Cellular Hematology Group (GFHC)"

_jcm, 2020, doi:10.3390/jcm9030808_

Round 1

Reviewer 1 Report

Comments:
In the present manuscript, Baccini et al review available techniques for platelet counting. Due to heterogeneous results among laboratory practices, the authors focus on interferences that could affect the platelet count and to detail the verification steps with minimal recommendations and to harmonize/standardize cellular hematology practices. They also propose solutions for several cases.  This review may be helpful for the specialists in laboratory medicine.

These are my main points.

An introduction to platelets is missing in the introduction section. It would be clearer if the review is first introduced with a ´Table of contents’. The readers will appreciate if the authors show images of cells obtained by different counting techniques in Section 1. Different counting techniques

There are many errors in the manuscript those need to be carefully checked, e.g.,

Line 91: Such decrease = Such a decrease Line 170-174: different font size Line 247 and others: platelets < 150x109/L = platelets < 150x109/L Line 162; 328, 409 and others: (clumps, fibrin, macroplatelets...) = (clumps, fibrin, macroplatelets, etc.) Section ` 3.3.1.1. Platelet clumps` was not well organized. The `bullet or dash’ symbols are not recommended to present in articles. etc.

Reviewer 2 Report

This is an excellent review of the assessment of platelet counts including normal, thrombocytopenia, and thrombocytosis. There are very many good parts of the report that could not easily be learned elsewhere. Overall, it is well-written and quite comprehensive.

However, there are a number of areas that could be improved as commented upon below.

There are several overall comments that may be repeated below in individual situations below.

One is that there needs to be more “clinical wisdom” inserted with the laboratory comments. A number of “errors” or uncertainties are the lack of appropriate clinical context to go with the laboratory discussion.

Second is another clinical issue. The authors bring up very rare issues to be “complete”. However, there are no estimates of frequency of many of them. Many are so rare they would be virtually reportable. The rarity needs to be emphasized and they should be described in proportion to how (in)frequent they are.

Third, platelet retics (IPF) are not taken very seriously. They are discussed but as if they have not been well-studied   

Finally the bibliography has a median reference date of 2006 or so = very early. There are 2 references from 2019 and several (? 3) from 2017 but almost all the rest are quite old with the clear majority earlier than 2010. That may represent the “state of the art” but it is also a call for more recent references in addition to or instead of the ones that are included.

The first paragraph refers to various techniques which are well-known to aficionados but not everyone else. They need to be described a little better even though there are more details provided later. Perhaps a figure of the different options could be included to illustrate similarities and differences including an explanation of why certain problems are better handled by one analytic method than another.

The authors describe platelet clumps in a quite comprehensive way. However certain issues are not optimally addressed:

What distinguishes real in vivo clumping from in vitro clumping eg EDTA pseudo-thrombocytopenia ? the authors only mention that bleeding signs and symptoms are not seen with in vitro clumping. The authors neglect that taking a finger stick and putting a drop of blood directly on a slide is also a good way to distinguish in vivo from in vitro clumping if the finger stick yields good flow. Not clear if this is what the authors mean later by “capillary sample” ? either way, this term needs clarification Many analyzers have statements in their manuals specifically prohibiting the use of tubes with other anti-coagulants than EDTA. Should these prohibitions be ignored if one suspects EDTA pseudo-thrombocytopenia ? The authors could explain why heparin is not used as the standard anticoagulant for blood counts. The authors say EDTA-dependent platelet antibodies cause clumping by activating platelet “tyrosine kinase”. At the very least, which tyrosine kinase is activated should be mentioned and which pathway is triggered by its activation would be useful as well. Causes of in vivo clumping, eg Von Willebrand IIB, should be mentioned and their diagnosis considered as a potential direction to resolve clumping. Overall, there are very many suggestions and possibilities considered. It would be helpful if a specific schema for clumping was suggested at the end so that it was clear which approach to use when. This would be a part of the figure later or stand alone since it is one of the commonest problems.

LARGE Platelets

“a_ _m_a_c_r_o_p_l_a_t_e_l_e_t_ _i_s_ _a_ _p_l_a_t_e_l_e_t_ _w_h_o_s_e_ _s_i_z_e_ _i_s_ _b_e_t_w_e_e_n_ _t_h_a_t_ _o_f_ _h_a_l_f_ _a_n_d_ _a_n_ _e_n_t_i_r_e_ _R_B_C_ _(_4_ _-_ _8_ _μm_)_._ _A_ _g_i_a_n_t_ _316 p_l_a_t_e_l_e_t_ _i_s_ _a_s_ _b_i_g_ _a_s_ _o_r_ _l_a_r_g_e_r_ _t_h_a_n_ _a_n_ _R_B_C_ _(_≥ _8_ _μm_)”

This classification/terminology seems arbitrary.  “Megathrombocyte” has also been used and defined either as at least 2/3 the size of an RBC or as larger than an RBC.  A better or wider description of terminology seems warranted including where the proposed terms are derived from. This reviewer is not clear that there exists a universally used and accepted terminology for very large platelet size. Some discussion on this is warranted.

“immune thrombocytopenic purpura” is not the correct term which is “immune thrombocytopenia” (see Rodeghiero Blood 2009 for standardization of terminology which has been in use since 2009)

Lines 335-336: “the relationship between MPV and the platelet count resembled or exceeded  the relationships found in normal subjects”.  This relationship has not been well defined or described in normals in this review.

The comment on the “smoothing of the histogram” re macroplatelets requires a figure with examples of what the authors mean.

The authors several times discuss confirming the platelet count with another method. It would be of interest to know how many laboratories (estimate per cent) have the ability to perform two different methods of platelet determination beyond manual counts. This is important because it is a frequent recommendation made by the article. Does this require two separate autoanalyzers ?  is this a plus for the Sysmex machines which can do optical and impedance measurements ?

“RBC examination can reveal schistocytes, dacryocytes, plasmodium”.  The authors include the previous quote in what to look for in RBC morphology. At the very least they need to include:  a) reticulocytes (polychromasia) in case of TTP, HUS, Evans syndrome; b) Howell-Jolly bodies which would indicate splenectomy which in turn would change the lower limit of the normal range (make it higher) and also eliminate the possibility of hypersplenism as a cause of thrombocytopenia; c) they mention plasmodium but not its “look alike” babesia; and d) dacrocytes especially are not well known (figures are needed to illustrate them and the other described findings as are done in figure 1 for “clumping”. Figure 3 goes partway but to truly be useful the other (complete set of) findings need inclusion. Also XLT is mentioned in the figure 3 legend but is not explained as being a form of Wiskott-Aldrich syndrome.  Finally the MYH9-RD figure does not clearly illustrate a dohle body (as it should), only a giant platelet.  

Lines 412-14: “An important point to highlight is that the rate of ITP is 2 - 4 cases per 100,000 person-years [70], which must be put forward upon reading the reported rate of inherited thrombocytopenia (IT) of 2.7 cases per 100,000 individuals as estimated in the Italian population [71]”. The incidence of ITP (immune thrombocytopenia) is considered to be 10 times as high as that of inherited thrombocytopenia not the same as it appears that this not completely clear sentence implies.

Lines 418-419: “Cases with mild to moderate thrombocytopenia can be misdiagnosed as ITP”.  This is true but IT cases can also have more severe thrombocytopenia and these cases can also be misdiagnosed as ITP.

Lines 421-22: “In patients with large platelets, the suspicion of MYH9-RD should be raised first since it is the most prevalent IT worldwide”.  While this may be true, there are a number of other types of macrothrombocytopenia with giant platelets so the focus should not be on just MYHY9-RD.

Also, the difference between ITP and MYH9-RD on smear is that there are too many too large platelets in the latter.

Information quantifying the MPV in a meaningful way has been limited by the variability among counters especially outside the limits of normal and at the limits of where the counters purpose to be accurate. This “conclusion” should either be included and substantiated or at least discussed with pros and cons. In severe thrombocytopenia, endothelial cell fragments often affect the MPV.

Lines 444-45: there should be an estimate of false positive thrombocytosis. It is likely there are 2 components. One is reporting a high platelet count when it is not really high; this seems likely to be very uncommon. More common is correctly discovering the count is high but having an inaccurate number.

There are several issues to consider in the discussion of thrombocytosis:

Estimates are needed of these occurrences especially cryoglobulinemia since it in general is very rare in this reviewer's experience How reliable are the “flags” in identifying platelet abnormalities in which the count is not correct----this can be true for all platelet counts Which types of leukemia are associated with small cell numbers or fragments ? it is likely that certain types are more prone to create this than are other types Red cell fragmentation is not well described in terms of diseases, frequency, which diseases are most likely to impact the platelet count Similarly the chance that there are sufficient bacteria to create a false elevation of the platelet count must be virtually reportable eg very rare

Lines 579-80:  “Rarely, thrombocytosis is primitive, non-clonal, and mostly with positive family history”. Familial thrombocytoses are usually mutations in TPO or c-mpl. Molecular testing is often possible.

Lines 582-594:  it might be best to reference some very recent articles on management of MPN instead of providing specific management in this one setting. For ET, it is not clear that all of the statements are correct.  Furthermore clinical management strategies are not provided elsewhere and do not seem to be an appropriate part of this laboratory review.  Diagnostic considerations applying to specific abnormalities "yes"; clinical management "no".

Round 2

Reviewer 1 Report

This version of the article is significantly improved as compared to the original article. The authors revised and added new figures accordingly. I have no major further comments. However, the authors need to check thoroughly the manuscript to correct other remaining errors.

Author Response

We have taken into account all the comments made by the referees and have modified the manuscript accordingly. The Fig.2a has been modified, the style has been harmonized and the suggested references have been added.

              We hope you will find that the current revised version is worthy of publication in the Journal of Clinical Medicine,             

Yours sincerely,
